# Rapidly Tuning the PID Controller Based on the Regional Surrogate Model Technique in the UAV Formation

**DOI:** 10.3390/e22050527

**Published:** 2020-05-06

**Authors:** Binglin Wang, Xiaojun Duan, Liang Yan, Juan Deng, Jiangtao Chen

**Affiliations:** 1College of Liberal Arts and Sciences, National University of Defense Technology, Changsha 410073, China; wangbinglin14@nudt.edu.cn (B.W.); xjduan@nudt.edu.cn (X.D.); dengjuan@amss.ac.cn (J.D.); 2China Aerodynamics Research and Development Center, Mianyang 621000, China; chenjt@cardc.cn

**Keywords:** surrogate model, proportional controller, UAV formation, classifier

## Abstract

The leader–follower structure is widely used in unmanned aerial vehicle formation. This paper adopts the proportional-integral-derivative (PID) and the linear quadratic regulator controllers to construct the leader–follower formation. Tuning the PID controllers is generally empirical; hence, various surrogate models have been introduced to identify more refined parameters with relatively lower cost. However, the construction of surrogate models faces the problem that the singular points may affect the accuracy, such that the global surrogate models may be invalid. Thus, to tune controllers quickly and accurately, the regional surrogate model technique (RSMT), based on analyzing the regional information entropy, is proposed. The proposed RSMT cooperates only with the successful samples to mitigate the effect of singular points along with a classifier screening failed samples. Implementing the RSMT with various kinds of surrogate models, this study evaluates the Pareto fronts of the original simulation model and the RSMT to compare their effectiveness. The results show that the RSMT can accurately reconstruct the simulation model. Compared with the global surrogate models, the RSMT reduces the run time of tuning PID controllers by one order of magnitude, and it improves the accuracy of surrogate models by dozens of orders of magnitude.

## 1. Introduction

The cooperative control of the unmanned aerial vehicle (UAV) formation is a research hotspot because of its widespread use, such as in forest fire surveillance, field surveillance, and antipoaching efforts [1,2]. Tuning controllers with efficient optimization methods is of prime importance to maintaining robust formation. In practice, the classical proportional-integral-derivative (PID) controller and its variations, such as the proportional controller and the proportional-integral controller, occupy 90% of industrial control [3]. However, many engineers think that many PID control loops in practice are not in high performance [3]. It is notable that the PID controller is parameter sensitive; hence, a more refined optimization method is required. Our study focuses on developing a high-efficient method to tune PID controllers.

Many researchers attempted to improve the robustness of UAVs through designing controllers. Some researchers researched the robustness of a single UAV. López-Estrada et al. [4] designed a robust fault detection and tracking controller system. Guzmán-Rabasa et al. [5] designed the fault detection and diagnosis system when a UAV was under partial or total actuator fault. The robust control of UAV formations has also attracted the attention of researchers. To design a robust UAV formation, which was independent of the environment, Viktor et al. [6] proposed an onboard relative localization method based on ultraviolet light. The robustness of UAVs with specific tasks has also been studied. Guerrero-Sánchez et al. [7] controlled single UAV with a cable-suspended payload through a hierarchical scheme with controllers based on energy and the linear matrix inequality. Tuning controllers plays an important role in keeping robust UAV systems.

To accelerate the process of tuning controllers, surrogate models (SUMOs) have been introduced because theoretical tuning methods or empirical tuning methods may be cumbersome or inefficient [8,9]. Regarding a system as a black box, SUMOs mimic relationships between system inputs and outputs. Hence, SUMOs have good adaptability. There are some common types of SUMOs, such as Kriging [10], polynomial chaos expansions (PCE) [11], polynomial chaos Kriging (PCK) [12], the radial basis function neural network (RBFNN) [13], and the generalized regression neural network (GRNN)[14]. It is worth noting that SUMOs have been widely used to optimize UAVs [15,16,17]. Researchers have used SUMOs to tune different controllers in different systems successfully, such as the mixing process [18], the cruise control system [19], and the unmanned underwater vehicle [20]. Among these systems, through offline optimization, various controllers were tuned, including the fuzzy logic controller [20], the proportional-integral controller [18], and the PID controller [18,19]. Additional SUMO-related techniques have been introduced. Lü [21] performed online optimization on high-purity distillation processes via the RBFNN. To investigate large, multidimensional input spaces, Matinnejad et al. [22] reduced the dimensionality of SUMOs, including the linear regression, the exponential regression, and the polynomial regression. Pan and Das [23] adopted Kriging to optimize the fractional order PID controller. Guerrero et al. [24] proposed a surrogate-based optimization workflow. Faruq et al. [25] proposed a Pareto-based surrogate modelling algorithm for optimizing PID controllers.

The previous works usually construct global SUMOs for control systems. However, the global SUMOs may not be the best choice for tuning controllers because researchers only concern the successful part of control systems. The accuracy of global SUMOs may be affected by the failed control results. For example, in this study, the UAV formation may generate singularity points when the control fails. Singularity points are fatal to the accuracy of SUMOs. In previous studies, singularity points did not raise close attention because researchers have prior experience, and successful samples were easy to be found [18,19,20,23,25]. The reasons for singularity points occurrence in this work are summarized as follows: First, the error of closed-loop systems may be reinforced compared to open-loop systems. Second, the solver in the simulation program may get exceptionally large values when the control fails. In this study, without prior experience, it is hard to avoid the singularity points before sampling. Therefore, recklessly tuning controllers using global SUMOs is problematic. A novel SUMO technique is thus needed to filter singularity points.

The remainder of this paper is presented as follows. Section 2 constructs the UAV formation simulation model and defines performance measures. In Section 3, the regional surrogate model technique (RSMT) is proposed based on the regional information entropy. In Section 4, the RSMT is used by different SUMOs, i.e., Kriging, PCE, PCK, the RBFNN, and the GRNN. Then, the Pareto fronts of the original simulation model and the RSMT are evaluated to compare their effectiveness.

## 2. The UAV Formation Model

### 2.1. The Leader–Follower Structure

Following Xu [26], fixed-wing UAVs form the UAV formation, which adopts the leader–follower (L–F) architecture: one leader leads the group while followers are controlled to maintain clearance between followers and the leader. The earth-fixed reference frame is built, and the dynamic models of UAVs [27] are given by
(1)$$\left\{\begin{array}{c} \begin{array}{c} {\dot{x}}_{L}=V_{L}\cos\phi_{L}\cos\theta_{L}\\ {\dot{y}}_{L}=V_{L}\sin\phi_{L}\cos\theta_{L}\\ {\dot{z}}_{L}=V_{L}\sin\theta_{L} \end{array}\\ \begin{array}{c} {\dot{x}}_{F}=V_{F}\cos\phi_{F}\cos\theta_{F}\\ {\dot{y}}_{F}=V_{F}\sin\phi_{F}\cos\theta_{F}\\ {\dot{z}}_{F}=V_{F}\sin\theta_{F} \end{array} \end{array}\right.,$$
where the subscripts L and F denote the leader and follower, respectively; *x*, *y*, and *z* denote the position of UAVs on the *x*-axis, *y*-axis, and *z*-axis; *V* is the forward velocity; θ is the track angle of UAVs; ω is the heading angular rate of UAVs, $\dot{\phi }=\omega$. As the angle between the forward direction and *x*-axis, the heading angle ϕ [26] can be given by
(2)$$\sin\phi =\frac{V_{y}}{\sqrt{V_{x}^{2}+V_{y}^{2}}},$$
where $V_{x}$ and $V_{y}$ are the components of *V* on the *x*-axis and *y*-axis. Because this paper focuses on fixed-wing UAVs which usually fly at the same height in a formation [26,28,29], we assume that UAVs do not change their height, i.e., $\theta_{L}=\theta_{F}=0$. Because the method of controlling all followers is identical, and there is no connection between followers, we examine only one follower instead of multiple followers. Figure 1 shows the geometry of the L-F structure in the x,y plane as follows:

The position relations between the leader and follower [26] are
(3)$$\left\{\begin{array}{c} \Delta f=\left(x_{L}-x_{F}\right)\cos\phi_{L}+\left(y_{L}-y_{F}\right)\sin\phi_{L}-f_{d}\\ \Delta l=-\left(x_{L}-x_{F}\right)\sin\phi_{L}+\left(y_{L}-y_{F}\right)\cos\phi_{L}-l_{d} \end{array}\right.,$$
where $f_{d}$ and $l_{d}$ are the desired forward and lateral clearances; Δf and Δl are the clearance errors in the forward and lateral directions.

The L-F structure aims to keep the desired clearance between the follower and the leader. The UAV formation is divided into the outer loop and the inner loop, which contain PID controllers and linear quadratic regulator (LQR) controllers, respectively. The outer loop controls the position dynamics to maintain the desired formation; the inner loop controls the UAV itself. The outer-loop controller generates commands into the inner-loop controller. The conceptual structure of the used UAV formation is shown in Figure 2. The reference generator gives the velocity and attitude of the leader [29]. Appendix A provides details of the inner-loop-controller design and the system matrices of a single UAV. Because the LQR controller belongs to optimum control, we only optimize the outer-loop controller, i.e., the PID controller, which is designed as follows.

### 2.2. Outer-Loop-Controller Design

It is assumed that *f* and *l* are the actual forward and lateral clearances from the leader reference frame [29]:(4)$$\left\{\begin{array}{c} f=\left(x_{L}-x_{F}\right)\cos\phi_{L}+\left(y_{L}-y_{F}\right)\sin\phi_{L}\\ l=-\left(x_{L}-x_{F}\right)\sin\phi_{L}+\left(y_{L}-y_{F}\right)\cos\phi_{L} \end{array}\right..$$
Differentiate the formula Equation (Equation 3) with respect to time, through substituting Equations (Equation 1) and (Equation 4), the rates of error change [29] are
(5)$$\left[\begin{array}{c} \Delta \dot{f}\\ \Delta \dot{l} \end{array}\right]=\left[\begin{array}{c} V_{L}-l\omega_{L}\\ -f\omega_{L} \end{array}\right]+\left[\begin{array}{c} -\cos\left(\phi_{F}-\phi_{L}\right)\\ -\sin\left(\phi_{F}-\phi_{L}\right) \end{array}\right]V_{F}.$$

The outer-loop controllers aim to generate proper commands, which will be tracked by the inner-loop controllers. We adopt two PID controllers as the outer-loop controllers in the forward and lateral directions. The two PID controllers are represented as $M_{l}$ and $M_{f}$, which are given as follows:(6)$$M_{l}\left(\Delta l\right)=K_{\mathrm{Pl}}\Delta l+K_{\mathrm{Il}}\int \Delta ldt+K_{\mathrm{Dl}}\frac{d\Delta l}{dt},$$
(7)$$M_{f}\left(\Delta f\right)=K_{\mathrm{Pf}}\Delta f+K_{\mathrm{If}}\int \Delta fdt+K_{\mathrm{Df}}\frac{d\Delta f}{dt},$$
where subscripts P,I,D represent the proportional gain, integral gain, and derivative gain of PID controllers, respectively; subscripts f and l represent the forward and lateral directions of UAVs, respectively. It is assumed that $K=\{K_{\mathrm{Pl}},K_{\mathrm{Il}},K_{\mathrm{Dl}},K_{\mathrm{Pf}},K_{\mathrm{If}},K_{\mathrm{Df}}\}$, which are user-defined and the key of tuning PID controllers. Then, Equation (Equation 5) can be written as
(8)$$\left[\begin{array}{c} \Delta \dot{f}\\ \Delta \dot{l} \end{array}\right]=\left[\begin{array}{c} V_{L}-l\omega_{L}\\ -f\omega_{L} \end{array}\right]+\left[\begin{array}{c} -\cos\left(\phi_{F}-\phi_{L}\right)\\ -\sin\left(\phi_{F}-\phi_{L}\right) \end{array}\right]V_{F}=\left[\begin{array}{c} -M_{f}\left(\Delta f\right)\\ -M_{l}\left(\Delta l\right) \end{array}\right],$$

Then, rearranging Equation (Equation 8), the following equation is gotten:(9)$$\left[\begin{array}{c} \cos\left(\phi_{F}-\phi_{L}\right)\\ \sin\left(\phi_{F}-\phi_{L}\right) \end{array}\right]V_{F}=\left[\begin{array}{c} M_{f}\left(\Delta f\right)+V_{L}-l\omega \\ M_{l}\left(\Delta l\right)-f\omega \end{array}\right].$$

Let $h_{F1}=M_{f}\left(\Delta f\right)+V_{L}-l\omega_{L}$, $h_{F2}=M_{l}\left(\Delta l\right)-f\omega_{L}$. The reference commands for the follower [29] are
(10)$$V_{F}^{r}=\sqrt{h_{F1}^{2}+h_{F2}^{2}},$$
(11)$$\phi_{F}^{r}=\left\{\begin{array}{c} \phi_{L}+\pi /2\\ \phi_{L}-\pi /2\\ \phi_{L}+\arctan\left(h_{F2}/h_{F1}\right)\\ \phi_{L}+\arctan\left(h_{F2}/h_{F1}\right)-\pi \\ \phi_{L}+\arctan\left(h_{F2}/h_{F1}\right)+\pi \end{array}\right.\begin{array}{c} \;h_{F1}=0,h_{F2}>0\\ \;h_{F1}=0,h_{F2}<0\\ \;h_{F1}>0\\ \;h_{F1}<0,h_{F2}\leq 0\\ \;h_{F2}<0,h_{F1}\geq 0 \end{array}.$$

### 2.3. Performance Measures of the UAV Formation

The follower’s trajectory generates response curves, whose horizontal axis is the time and whose vertical axis is the clearance to the leader in two directions. Response curves are evaluated via three kinds of commonly used measures, as follows:**Steady-state value** ($y_{\mathrm{st}}$): the stable value of the response curve, which is the direct aim of the controller.**Overshoot** (σ): the maximum peak value of the response curve measured from the desired response, which is given by [30]
(12)$$\sigma \%=\frac{y_{\max}-y_{\mathrm{st}}}{y_{\mathrm{st}}}\times 100\%,$$
where $y_{\max}$ is the peak value of the response curve beyond $y_{\mathrm{st}}$.**Accommodation time** ($t_{a}$): the time at which the response curve enters a specific interval around the desired response and no longer exceed the specific interval.

The lateral and forward motion are mutually independent and controlled by different controllers, so $y_{\mathrm{st}}$ and σ are divided into the lateral steady-state value $l_{\mathrm{st}}$, the forward steady-state value $f_{\mathrm{st}}$, the lateral overshoot $\sigma_{l}$, and the forward overshoot $\sigma_{f}$.

## 3. The Regional Surrogate Model Technique Based on the Regional Information Entropy

A change of systems, especially for actual physical systems, is usually a gradual process, which makes the response surface smooth, such as in computational fluid dynamics [31], aerology [32], and hydrology [33]. Thus, the global SUMO is adopted in most cases. However, in this study, singular points make the response surface rough, and the global SUMO is no longer effective. There are two reasons for this phenomenon: First, the UAV formation is a closed-loop system, which may reinforce errors. Second, the solvers in simulation fail to solve equations, which lead to the generation of singular points. Without prior experience for determining the selection of parameter space, singular points are unavoidable, and it is essential to mitigate the effect caused by singular points. Based on the regional information entropy, the RSMT is proposed as a means of reconstructing the UAV formation.

### 3.1. Regional Information Entropy Analysis

The SUMO can be viewed as a way to reconstruct the information of systems. Hence, a reasonable SUMO should fully display useful information and avoid interference from useless information, which in this study is mainly caused by singular points. Hence, analyzing the regional information entropy relationship can provide us with a decision basis for screening information. As a way of measuring the information content, information entropy *S* [34] is given by
(13)$$S=-\int p\left(x\right)\ln p\left(x\right)dx,$$
where *x* is the output of the system and $p\left(x\right)$ is the probability distribution function (PDF) of *x*. There is a positive correlation between *S* and information content.

For simplicity, we examine only one input with one output. The space of success (SOS) is the success interval $I_{\mathrm{succ}}$, which is the set of successful outputs. $I_{\mathrm{succ}}$ needs to contain all potential optimal solutions. The space of failure (SOF), i.e., the failed interval $I_{\mathrm{fail}}$, is the set of failed outputs, and $I_{\mathrm{succ}}\cap I_{\mathrm{fail}}=Ø$. $P_{\mathrm{fail}}$ and $P_{\mathrm{succ}}$ are the probabilities of outputs belonging to $I_{\mathrm{fail}}$ and $I_{\mathrm{succ}}$ respectively. Because success and failure are complementary events, $P_{\mathrm{succ}}+P_{\mathrm{fail}}=1$. $S_{\mathrm{succ}}$ and $S_{\mathrm{fail}}$ are the information entropy of $I_{\mathrm{succ}}$ and $I_{\mathrm{fail}}$, respectively. Containing useful and useless information, the entropy of the entire system is $S_{\mathrm{succ}}+S_{\mathrm{fail}}$, which is the whole information entropy of the global SUMO, i.e., the global SUMO completely reconstruct the entire system. The information entropy ratio of two kinds of information is *W*, $W=S_{\mathrm{succ}}/S_{\mathrm{fail}}$. It is assumed that $I_{\mathrm{succ}}$ and $I_{\mathrm{fail}}$ are both uniform distributions; then, the PDF of *x* is given by
(14)$$p\left(x\right)=\left\{\begin{array}{cc} P_{\mathrm{succ}}/\left(b-a\right) & a\leq x\leq b\\ P_{\mathrm{fail}}/\left(a-x_{\min}+x_{\max}-b\right) & x_{\min}\leq x<a\;\mathrm{or}\;b<x\leq x_{\max} \end{array},\right.$$
where *a*, *b* are the bounds of $I_{\mathrm{succ}}$, $I_{\mathrm{succ}}\in \left[a,b\right]$; $x_{\min}$, $x_{\max}$ are the lower limit and upper limit of *x*, $I_{\mathrm{fail}}\in \left[x_{\min},a\right)\cup \left(b,x_{\max}\right]$. $S_{\mathrm{fail}}$ and $S_{\mathrm{succ}}$ are given by
(15)$$\begin{array}{cc} S_{\mathrm{fail}} & =-\int_{x_{\min}}^{a}\frac{P_{\mathrm{fail}}}{a-x_{\min}+x_{\max}-b}\ln\frac{P_{\mathrm{fail}}}{a-x_{\min}+x_{\max}-b}dx\\ \; & -\int_{b}^{x_{\max}}\frac{P_{\mathrm{fail}}}{a-x_{\min}+x_{\max}-b}\ln\frac{P_{\mathrm{fail}}}{a-x_{\min}+x_{\max}-b}dx\\ \; & =-P_{\mathrm{fail}}\ln\frac{P_{\mathrm{fail}}}{a-x_{\min}+x_{\max}-b}, \end{array}$$
(16)$$S_{\mathrm{succ}}=-\int_{a}^{b}\frac{P_{\mathrm{succ}}}{b-a}\ln\frac{P_{\mathrm{succ}}}{b-a}dx=-P_{\mathrm{succ}}\ln\frac{P_{\mathrm{succ}}}{b-a}.$$

It is assumed that a=−5, b=5, $x_{\min}=-1000$, and $x_{\max}=1000$. According to Equations (Equation 15) and (Equation 16), Case 1 in Figure 3 shows the relationship between $P_{\mathrm{fail}}$ and *W*. If there is no prior experience in parameter selection, $P_{\mathrm{succ}}$ will be small, which makes $S_{\mathrm{fail}}>S_{\mathrm{succ}}$; in other words, useless information covers up useful information. Hence, the new SUMO technique should prevent useful information from being concealed by increasing *W*. In practice, we do not consider the output value and input parameter of failed results, which is the source of useless information. Hence, ignoring the difference within failed results is reasonable. Regarding failed results as one event, $S_{\mathrm{fail}}$ and $S_{\mathrm{succ}}$ can be given by
(17)$$S_{\mathrm{fail}}=-P_{\mathrm{fail}}\ln\left(P_{\mathrm{fail}}\right),$$
(18)$$S_{\mathrm{succ}}=-\int_{a}^{b}p_{\mathrm{succ}}\left(x\right)\ln p_{\mathrm{succ}}\left(x\right)dx,$$
where $p_{\mathrm{succ}}\left(x\right)$ is the PDF of $I_{\mathrm{succ}}$, $x\in I_{\mathrm{succ}}$; $P_{\mathrm{succ}}=\int_{a}^{b}p_{\mathrm{succ}}\left(x\right)dx$. Assuming that $I_{\mathrm{succ}}$ is the uniform distribution, $p_{\mathrm{succ}}\left(x\right)$ and *W* are given by
(19)$$p_{\mathrm{succ}}\left(x\right)=\left\{\begin{array}{cc} 0 & x<a\;\mathrm{or}\;x>b\\ P_{\mathrm{succ}}/\left(b-a\right) & a\leq x\leq b \end{array},\right.$$
(20)$$W=\frac{S_{\mathrm{succ}}}{S_{\mathrm{fail}}}=\frac{-P_{\mathrm{succ}}\ln\left(P_{\mathrm{succ}}/\left(b-a\right)\right)}{-P_{\mathrm{fail}}\ln P_{\mathrm{fail}}}=\frac{\left(1-P_{\mathrm{fail}}\right)\ln\left(\left(1-P_{\mathrm{fail}}\right)/\left(b-a\right)\right)}{P_{\mathrm{fail}}\ln P_{\mathrm{fail}}}.$$

Case 2 in Figure 3 shows the relationship between $P_{\mathrm{fail}}$ and *W*. For Case 2 in Figure 3, *W* is always larger than 5, which means that the proportion of useless information is reduced, and useful information constitutes almost the entirety of the information. Ignoring the difference within $I_{\mathrm{fail}}$ effectively eliminates useless-information interference. Moreover, to verify the results shown Case 1 in Figure 3, Case 3 in Figure 3 shows the same relation when the distribution of *x* is Student’s *t*-distribution. A detailed discussion is provided in Appendix B. The values of *a*, *b*, $x_{\min}$, and $x_{\max}$ impact *W* slightly; hence, the changes of these values do not affect the related conclusions.

In conclusion, constructing SUMOs needs to reduce the concealing of useful information. Different results should be differently treated according to the aim of constructing SUMOs. Based on the analysis presented above, the RSMT is proposed as a means of tuning PID controllers in the UAV formation.

### 3.2. The Regional Surrogate Model Technique

Section 3.1 discusses the relationship between $S_{\mathrm{fail}}$ and $S_{\mathrm{succ}}$. To purify information, we propose the RSMT, which is shown in Algorithm 1 and Figure 4 (the source code can be obtained from the authors). Class 1 means that the sample belongs to the SOS, and class 0 is contrary to class 1. Whether the control is successful or failed is determined according to user-defined thresholds. As discussed in Section 3.1, we ignore the difference within $I_{\mathrm{fail}}$ and focus on $I_{\mathrm{succ}}$. Instead of the global SUMO, the RSMT constructs the regional SUMO, which reconstructs the system only in the SOS. The RSMT can also be viewed as a weighted global SUMO: the weight of training samples belonging to the SOS is one; the weight of other samples is zero.
**Algorithm 1** The regional surrogate model technique.**Input:** the number of initial samples *N*; the parameter space PS; the criteria of the SOS.**Output:** A classifier; a regional SUMO **Definition**: the selected training set for the SUMO ST; the training set for classifier CT1: Make the initial sample selection from the PS and get *N* samples2: Put selected samples into the simulation model to get their response3: **for** each sample and its response4:  **if**
*i*-th sample belongs to the SOS5:   add *i*-th sample and its response into ST;6:   classify *i*-th sample with class 1;7:   add *i*-th sample and its class into CT;8:  **else**9:    classify *i*-th sample with class 0;10:   add *i*-th sample and its class into CT11:  **end if**12: **end for**13: Train the SUMO by ST14: Train the classifier by CT

In the RSMT, distinguishing class 0/1 requires user-defined thresholds, which should be more lenient than control objectives to avoid ignoring potential optimal solutions. After classing samples in accordance with thresholds, a classifier is trained by samples and their subordinate class to find the boundary of the SOS, which is difficult to describe analytically. Only samples belonging to the SOS are selected as the training set of SUMOs. Thus, it is limited to the use of trained SUMOs, whose use process is shown in Figure 5. When inputting new parameters, the first step is classifying the new parameters by the trained classifier. If it can lead to successful control, the outputs of the parameters are predicted by the trained SUMOs. Otherwise, these parameters are abandoned because they do not belong to potential optimal solutions.

Instead of optimizing parameters of SUMOs, the RSMT focuses on selecting a more reasonable training set for constructing SUMOs in a specific region. In this paper, the classifier adopts a decision tree, which performs well in binary classification and is given in Algorithm 2, following [35]. In a sense, the global SUMO is the combination of multiple regional SUMOs, whose marginal values are the same. If the response surface is rough, it is difficult to mimic the dramatically changed response surface using only one SUMO. Because the RSMT constructs the regional SUMO in the SOS whose outputs are limited, the selected response surface will be smooth, as a result of which the regional SUMO has high accuracy without missing the potential optimal solution.
**Algorithm 2** Generating decision tree.**Input:***D*: the training set; *C*: the attribute set.**Output:** A decision tree **Function** TreeGenerate (D,C)1: Create a node *N*2: **if** tuples in *D* belong to only one class *C*
**then**3:  label *N* as a leaf node with class *C*; **return**4: **end if**5: **if**
*C* is empty **OR** the samples of *D* are of the same class **then**6:  set label *N* as the leaf node with the most common class in *D*; **return**7: **end if**8: Find the best splitting criterion $c_{*}$ from *C*9: **for** each $c_{*}$ do10:  add a branch below *N*, corresponding to $c_{*}=c_{*}^{v}$11:  $D_{v}$ is the subset of *D* with $c_{*}=c_{*}^{v}$12:   **if**
$D_{v}$ is empty **then**13:   label the branch node as the leaf node with the most common class in *D*; **return**14:   **else**15:   set TreeGenerate $\left(D_{v},C\setminus \left\{c_{*}\right\}\right)$ as the branch node16:   **end if**17: **end for**

## 4. Simulation and Results

### 4.1. Evaluation Results for SUMOs Based on the RSMT

In this study, we attempt to substitute the SUMO for the UAV formation in Section 2. As parameters to be optimized, inputs are $K=\{K_{\mathrm{Pl}},K_{\mathrm{Il}},K_{\mathrm{Dl}},K_{\mathrm{Pf}},K_{\mathrm{If}},K_{\mathrm{Df}}\}$. With no correlation between them, the six intervals of K form the entire parameter space. Outputs are five performance measures, i.e., $l_{\mathrm{st}}$, $f_{\mathrm{st}}$, $\sigma_{l}$, $\sigma_{f}$, and $t_{a}$. The trained SUMO is evaluated using the root mean squared error (RMSE) [36], which is given by
(21)$$RMSE=\sqrt{\frac{1}{n_{t}}\sum_{i=1}^{n_{t}}(y_{i}-{\hat{y_{i}})}^{2}},$$
where $n_{t}$ is the number of test points; $\hat{y_{i}}$ and $y_{i}$ are the estimated value and exact value of the *i*th test point, respectively.

At first, the initial sample selection adopts Latin hypercube sampling. Table 1 shows the evolution results regarding whether or not SUMOs are constructed through the RSMT, and Appendix C provides a brief introduction to the applied SUMOs, including Kriging, PCE, PCK, the RBFNN, and the GRNN. If the RSMT is not adopted, global SUMOs are constructed. In the simulation, the initial positions of the leader and the follower are (200m,200m) and (0,400m), which are the same as [29]. Control objectives are $f_{d}=100\;m$ and $l_{d}=-100\;m$, which are also the same as [29]. Optimization aims to find the optimal PID controllers that can maintain the formation with low $l_{\mathrm{st}}$, $f_{\mathrm{st}}$, $\sigma_{l}$, $\sigma_{f}$, and $t_{a}$ under constraints that $\left|\Delta f\right|<3\%\left|f_{d}\right|=3\;m$ and $\left|\Delta l\right|<3\%\left|l_{d}\right|=3\;m$. Minimizing $\sigma_{l}$ and $\sigma_{f}$ aims to reduce the risk of UAVs collisions, which are important in the UAV formation [37]. In this work, the thresholds for the SOS are $\pm f_{d}$ and $\pm l_{d}$, which mean that the regional SUMO will be constructed in the region $\left|\Delta f\right|<100\;m$ and $\left|\Delta l\right|<100\;m$.

In Table 1, the corresponding values of each SUMO are the RMSE of $f_{\mathrm{st}}$. “Time” is the total run time of constructing all SUMOs. It is assumed that the intervals of six inputs are the same. In Table 1, the intervals of K are the intervals of six inputs, i.e., $K_{\mathrm{Pl}}$, $K_{\mathrm{Il}}$, $K_{\mathrm{Dl}}$, $K_{\mathrm{Pf}}$, $K_{\mathrm{If}}$, and $K_{\mathrm{Df}}$. For an interval of K, the first row shows results for regional SUMOs through the RSMT, and the second row shows results for global SUMOs. Regional SUMOs generate from our proposed method, i.e., the RSMT. Meanwhile, global SUMOs are the traditional way to construct SUMOs, i.e., adopting all samples to construct SUMOs. Because the second row of each K constructs the global SUMO without the classifier, the result of “Classification accuracy” is “null”. The calculation condition is MATLAB 2019a, Intel (R) Xeon (R) W-2145 CPU @ 3.70GHz, 32GB Memory, Windows 10.

According to Table 1, trained regional SUMOs are significantly better than trained global SUMOs. This phenomenon is in accord with the information relationship in Section 3.1. Adopting the same calculation method in Section 3.1, Table 2 shows the information entropy ratio *W* of corresponding K in Table 1. Because the sample size is limited, we use the frequency approximation as the probability. $W_{1}$ is the information entropy ratio with the RSMT and is corresponding to the first row of each K in Table 1. $W_{1}$ is large with the RSMT, which reserves useful information and avoids useless-information interference. At the same time, $W_{2}$ is the information entropy ratio without the RSMT and is corresponding to the second row of each K in Table 1. $W_{2}$ is relatively small and will decrease with the decrease of $P_{\mathrm{succ}}$. It means that useful information will be covered by useless information with the decrease of $P_{\mathrm{succ}}$. Moreover, $W_{1}$ is always larger than $W_{2}$. $W_{1}/W_{2}$ increases quickly with the expansion of K. $W_{1}/W_{2}$ shows the change of the information entropy ratio with the RSMT or not. The RSMT effectively increases the proportion of useful information. Figure 6 shows the relationship between $W_{1}/W_{2}$ and the effects of the RSMT, which is represented by orders of magnitude change of Kriging’s RMSE. The increase of $W_{1}/W_{2}$ brings the better effect of the RSMT, especially when $W_{1}/W_{2}$ is relatively small. The RSMT increases the proportion of useful information entropy, which leads to accurate regional SUMOs.

Regarding the computational cost, according to Table 1, training regional SUMOs is more time-saving with the expansion of K because the number of successful parameters will be fewer when the whole parameter space is larger. There are five different types of SUMOs, and the RSMT performs well in each of them, which shows that the RSMT has good generalization ability and is essential to various SUMOs.

In conclusion, the RSMT successfully filters singular points and maintains the high accuracy and low computational cost of regional SUMOs. In the next subsection, the optimal parameters of PID controllers are found through the RSMT.

### 4.2. Tuning PID Controllers Through the RSMT

We try to tune two PID controllers whose six inputs all belong to $\left[0,0.3\right]$. In the simulation, white noise is added to the lateral and forward positions of the leader, and the noise energy of white noise is $1\times 10^{-2}$. As mentioned above, class 1 indicates that corresponding samples belong to the SOS, and the meaning of class 0 is reversed: class 0 indicates that corresponding samples belong to the SOF. The global Sobol sensitivity analysis is adopted to analyze the relationships between inputs and class 0/1, whose results are shown in Figure 7. According to Figure 7, $K_{\mathrm{Pl}}$ and $K_{\mathrm{Il}}$ are the most important inputs. Figure 8 shows prediction results of trained classifier in the $K_{\mathrm{Pl}}$, $K_{\mathrm{Il}}$ plane. In Figure 8, blue symbols mean that the corresponding sample belongs to class 0, i.e., the SOF, and red symbols mean that the corresponding sample belongs to class 1, i.e., the SOS. “Correct” and “Incorrect” are the correctness of the classifier’s prediction. The classification accuracy of the trained decision tree is 84.0%. Figure 9 shows the receiver operating characteristic (ROC) curve of the trained classifier. The area under the ROC curve equals 0.88. The false positive rate is 14%, and the true positive rate is 84%. Hence, the trained classifier is accurate and reliable.

Table 3 presents the RMSEs of five performance measures by trained regional SUMOs through the RSMT. During the MATLAB/SIMULINK simulation, the solver is ode1 (Euler), and fixed-step size is $1\times 10^{-3}$, so $t_{a}$ is represented by the step number. Kriging, PCE, and PCK perform better than the RBFNN and GRNN. Adopting Kriging to tune PID controllers, the MATLAB function “paretosearch” and “gamultiobj” are used to find Pareto fronts. The run times of the simulation model and regional Kriging are denoted by “Simulation model time” and “Regional Kriging time”, respectively, in Table 4. The process of optimization is greatly accelerated. The cost of searching by Kriging is substantially lower than that of searching by the simulation model.

We examine solutions of “paretosearch” which meets the constraint conditions of optimization mentioned in Section 4.1, i.e., $\left|{\Delta f}_{\mathrm{st}}\right|<3m$ and $\left|{\Delta l}_{\mathrm{st}}\right|<3m$. Selected Pareto solutions are evaluated using the technique for order of preference by similarity to ideal solution (TOPSIS) [38]. The solutions of Kriging are also entered into the simulation model to obtain results for evaluation. Then, simulation results from different sources are evaluated by TOPSIS. Table 5 shows the Pareto solutions of different sources. According to Table 5, the scores of the two sources are similar to each other. It means that optimal solutions of regional Kriging are also able to be optimal solutions of the simulation model. Adopting the solution with highest score, i.e., $\left[0.300,0.0001,0.300,0.291,0.164,0.145\right]$, Figure 10 shows the trajectories of UAVs formation when the heading angle of the leader $\phi_{L}$ changes according to the sine function. When the heading angle of the leader UAV changes, the follower UAV can timely adjust to maintain the formation.

With the RSMT, regional Kriging accurately replaces the simulation model to find optimal solutions. Abandoning useless information does not affect searching optimal parameters. The RSMT can accelerate the optimization process with high accuracy and low computational time simultaneously.

## 5. Conclusion and Discussion

To accelerate the process of tuning PID controllers, this work proposes the RSMT based on analyzing the regional information entropy relationship. The RSMT discards redundant information to construct the regional SUMO. A classifier is introduced to define the boundary of the regional SUMO. According to calculation results, the RSMT significantly improves the accuracy of SUMOs and reduces computational expense. The results verify the theoretical analysis of the regional information entropy relationship. To corroborate the reliability of the RSMT, the Pareto fronts are searched by regional SUMOs and the simulation model, respectively. It is found that different Pareto fronts are similar to each other. The RSMT reduces the run time of parameter optimization by one order of magnitude, and it gets reliable optimization results.

The RSMT can tune PID controllers with high efficiency and accuracy, and be available for various types of SUMOs. In the process of tuning PID controllers, the RSMT significantly reduces the singular-point interference, improves the accuracy of SUMOs, and reduces computational expense. Not only limited optimization of the UAV formation, but the RSMT can also be extended for tuning PID controllers in various systems because SUMOs only concern inputs and outputs of systems. In future research, we prone to solve the application problem of the RSMT in high-dimensional situations, which may be solved by combining sequential sampling and dimensionality reduction technology.

## Figures and Tables

**Figure 1 entropy-22-00527-f001:**
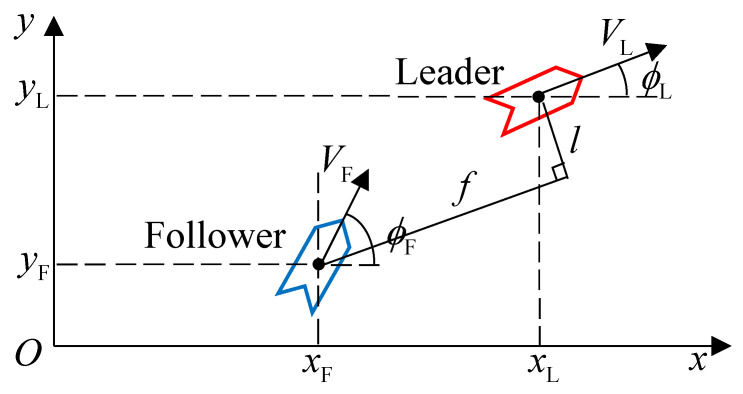
The leader–follower structure [29] in the x,y plane. One leader leads the group while the follower is controlled to maintain clearance between the follower and the leader.

**Figure 2 entropy-22-00527-f002:**
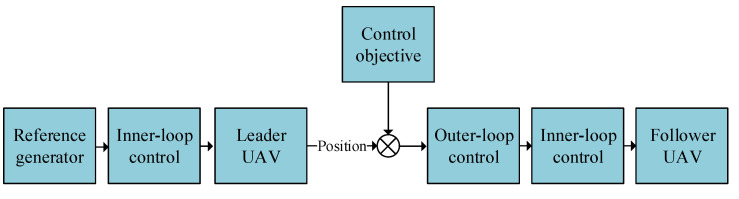
The conceptual block diagram of the leader–follower UAV formation.

**Figure 3 entropy-22-00527-f003:**
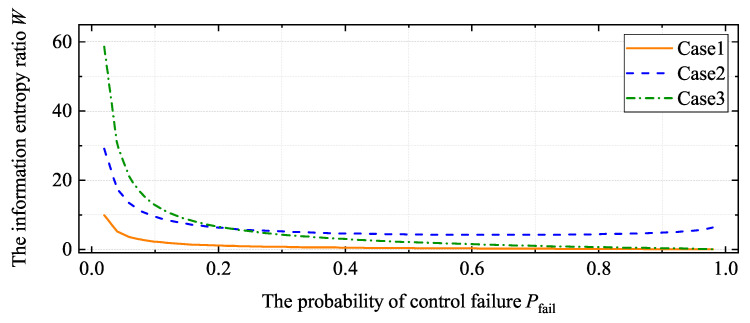
The relationship between $P_{\mathrm{fail}}$ and *W*. Case 1: all samples are used in the SUMO construction. $I_{\mathrm{succ}}$ and $I_{\mathrm{fail}}$ are both uniform distributions. Case 2: the failed results are viewed as one event, and $I_{\mathrm{succ}}$ is the uniform distribution. Case 3: $I_{\mathrm{succ}}$ is the uniform distribution and $I_{\mathrm{fail}}$ is the *t*-distribution. Filtering useless information is essential for preventing useful information from being submerged. Ignoring the difference within $I_{\mathrm{fail}}$ effectively eliminates useless-information interference.

**Figure 4 entropy-22-00527-f004:**
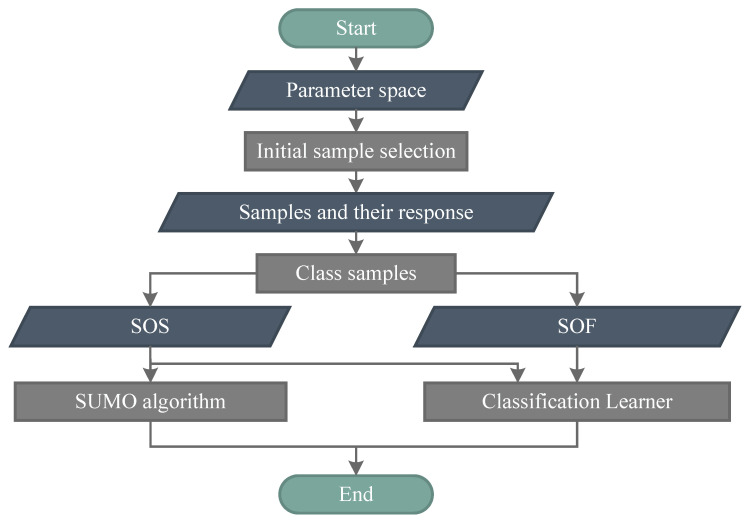
The regional surrogate model technique. The SUMO is constructed only in the SOS, whose boundary is found by a classification learner.

**Figure 5 entropy-22-00527-f005:**
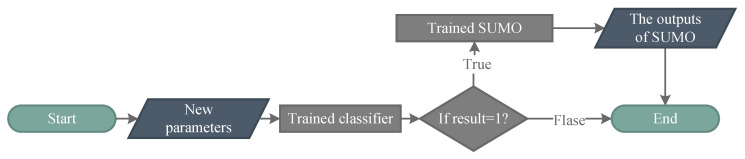
The use of trained SUMOs obtained by the RSMT. Inputs are judged by the classifier, and only the inputs belonging to the SOS are predicted by the trained SUMO. If the result of classifier equals to 1, it means that the new parameters belong to the SOS.

**Figure 6 entropy-22-00527-f006:**
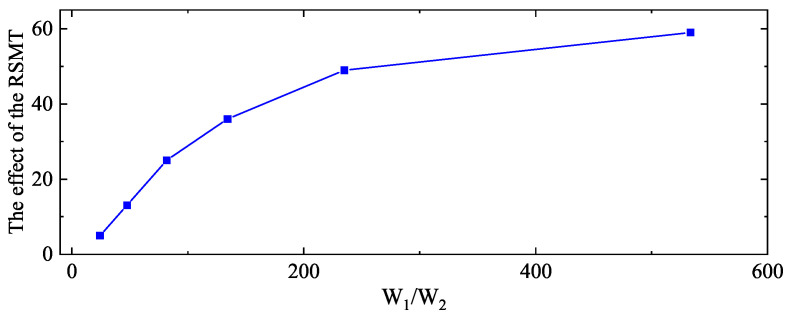
The relationship between $W_{1}/W_{2}$ and the effects of the RSMT. $W_{1}/W_{2}$ shows the change of the information entropy ratio with the RSMT or not. Effects of the RSMT are represented by orders of magnitude change of Kriging’s RMSE in Table 1. The increase of $W_{1}/W_{2}$ brings the better effect of the RSMT, especially when $W_{1}/W_{2}$ is limited.

**Figure 7 entropy-22-00527-f007:**
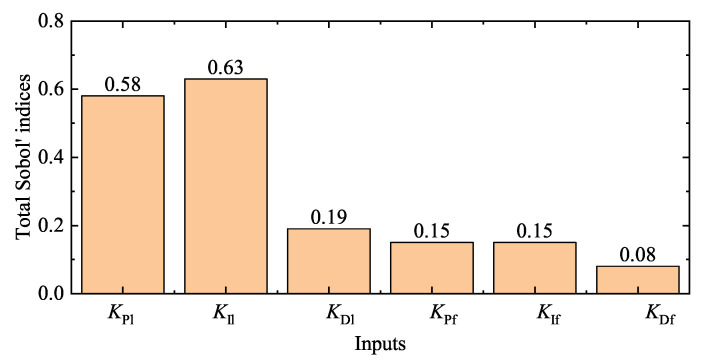
Total Sobol’ indices. $K_{\mathrm{Pl}}$ and $K_{\mathrm{Il}}$ are the most important inputs which effect classification results of inputs.

**Figure 8 entropy-22-00527-f008:**
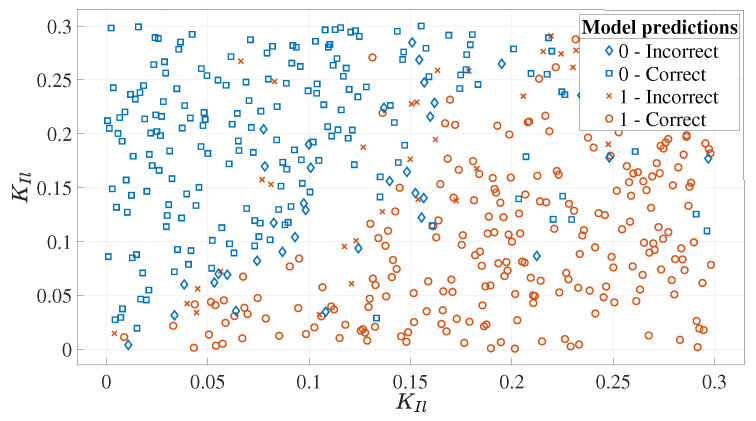
The prediction results of trained classifier in the $K_{\mathrm{Pl}}$, $K_{\mathrm{Il}}$ plane. Blue symbols mean that the corresponding sample belongs to class 0, and red symbols mean that the corresponding sample belongs to class 1. Class 1 indicates that corresponding samples belongs to the SOS, and the meaning of class 0 is reversed. The classification accuracy of the trained classifier is 84%.

**Figure 9 entropy-22-00527-f009:**
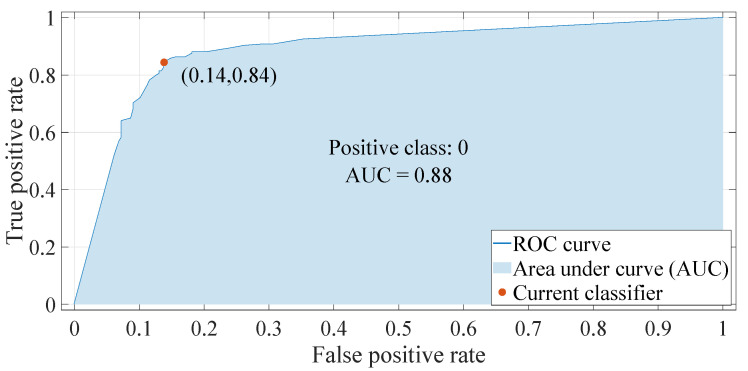
The receiver operating characteristic (ROC) curve of the trained classifier. The area under curve equals to 0.88. The false positive rate is 14% and the true positive rate is 84%. The trained classifier is accurate and reliable.

**Figure 10 entropy-22-00527-f010:**
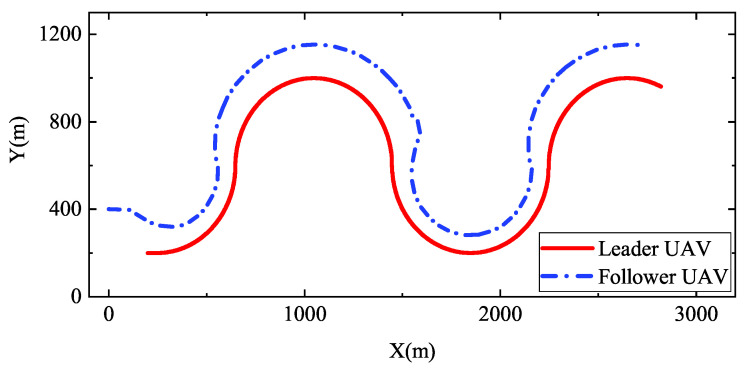
Trajectories of UAVs formation. The heading angle of the leader $\phi_{L}$ changes according to the sine function. PID controllers adopt the solution with the highest score. i.e., $K\;=\;\left[0.300,0.0001,0.300,0.291,0.164,0.145\right]$

**Table 1 entropy-22-00527-t001:** Comparison of regional and global SUMOs for $f_{\mathrm{st}}$.

*K* ${}^{a}$	Classification Accuracy (%) ${}^{b}$	Time (s) ${}^{c}$	Kriging ${}^{d}$	PCE ${}^{d}$	PCK ${}^{d}$	GRNN ${}^{d}$	RBFNN ${}^{d}$
$\left[0,0.1\right]$	85	350.7	8.3	12.3	7.7	21.0	14.9
	null	437.6	$1.62\times 10^{6}$	$1.41\times 10^{6}$	$1.37\times 10^{6}$	$1.26\times 10^{6}$	$1.63\times 10^{7}$
$\left[0,0.2\right]$	85	317.9	11.4	16.2	16.3	20.8	26.6
	null	289.6	$8.20\times 10^{14}$	$1.35\times 10^{14}$	$2.08\times 10^{14}$	$8.11\times 10^{13}$	$1.90\times 10^{15}$
$\left[0,0.3\right]$	84.6	318.3	8.6	9.2	8.9	14.2	29.4
	null	567.7	$9.41\times 10^{25}$	$2.71\times 10^{25}$	$3.23\times 10^{26}$	$1.97\times 10^{25}$	$5.27\times 10^{26}$
$\left[0,0.4\right]$	81.8	307.4	7.4	9.4	8.1	11.5	16.5
	null	322.5	$3.76\times 10^{37}$	$3.32\times 10^{36}$	$6.40\times 10^{37}$	$4.43\times 10^{36}$	$1.40\times 10^{38}$
$\left[0,0.5\right]$	79	111.5	7.7	8.6	7.4	12.2	12.3
	null	312.3	$4.63\times 10^{51}$	$7.20\times 10^{50}$	$4.44\times 10^{50}$	$5.53\times 10^{50}$	$1.99\times 10^{52}$
$\left[0,0.6\right]$	80.2	88.5	7.4	8.7	8.5	12.1	11.4
	null	356.4	$8.21\times 10^{59}$	$3.94\times 10^{59}$	$3.94\times 10^{59}$	$4.68\times 10^{59}$	$1.09\times 10^{61}$

${}^{a}$ Two PID controllers have six inputs which share the same interval, and inputs are $K=\{K_{\mathrm{Pl}},K_{\mathrm{Il}},K_{\mathrm{Dl}},K_{\mathrm{Pf}},K_{\mathrm{If}},K_{\mathrm{Df}}\}$. For a value of K, the first and second rows show results for regional SUMOs and global SUMOs, respectively. ${}^{b}$ Because the second row of each K constructs the global SUMO without the classifier, the result of “Classification accuracy” is “null”. ${}^{c}$ “Time” is the total run time for constructing all SUMOs.  ${}^{d}$ Values of SUMOs represent the RMSE of $f_{\mathrm{st}}$ for trained SUMOs. The RSMT can significantly reduce errors and save computation time.

**Table 2 entropy-22-00527-t002:** The information entropy ratio *W* in actual computation.

*K*	$f_{\mathrm{st},\max}$	$f_{\mathrm{st},\min}$	$P_{\mathrm{fail}}$	$P_{\mathrm{succ}}$	$W_{1}$ ${}^{a}$	$W_{2}$ ${}^{b}$	$W_{1}/W_{2}$
$\left[0,0.1\right]$	$3.14\times 10^{2}$	$-6.81\times 10^{8}$	0.41	0.59	9.36	0.39	24.06
$\left[0,0.2\right]$	$1.18\times 10^{12}$	$-5.35\times 10^{16}$	0.44	0.56	9.13	0.19	47.66
$\left[0,0.3\right]$	$3.26\times 10^{23}$	$-1.35\times 10^{28}$	0.45	0.55	9.04	0.11	81.68
$\left[0,0.4\right]$	$1.26\times 10^{25}$	$-1.62\times 10^{39}$	0.51	0.49	8.59	0.06	134.23
$\left[0,0.5\right]$	$1.76\times 10^{26}$	$-3.46\times 10^{53}$	0.59	0.41	8.15	0.03	234.94
$\left[0,0.6\right]$	$4.03\times 10^{36}$	$-1.68\times 10^{62}$	0.76	0.24	7.74	0.01	533.58

${}^{a}$$W_{1}$ is the information entropy ratio with the RSMT and is corresponding to the first row of each K in Table 1. ${}^{b}$
$W_{2}$ is the information entropy ratio without the RSMT and is corresponding to the second row of each K in Table 1. $W_{1}/W_{2}$ increases quickly with the expansion of K. Without the RSMT, useless information will cover up useful information. The RSMT effectively increases the proportion of useful information.

**Table 3 entropy-22-00527-t003:** The accuracy of regional SUMOs through the RSMT in $K\in \left[0,0.3\right]$.

	$\sigma_{f}{}^{a}$	$\sigma_{l}{}^{a}$	$f_{\mathrm{st}}{}^{a}$	$l_{\mathrm{st}}{}^{a}$	$t_{a}{}^{a}$
Kriging	5.41	19.82	16.80	26.59	$1.63\times 10^{4}$
PCE	5.65	21.52	17.28	32.23	$2.11\times 10^{4}$
PCK	5.84	17.77	19.51	31.25	$1.73\times 10^{4}$
GRNN	15.15	65.43	15.74	28.88	$3.26\times 10^{4}$
RBFNN	18.84	37.34	37.33	155.45	$6.36\times 10^{4}$

${}^{a}$$\sigma_{f}$, $\sigma_{l}$, $f_{\mathrm{st}}$, $l_{\mathrm{st}}$ and $t_{a}$ denote the RMSE of them getting from Kriging, PCE, PCK, the RBFNN, and the GRNN, respectively. Every SUMO is accurate, but Kriging, PCE, and PCK perform better than the RBFNN and GRNN.

**Table 4 entropy-22-00527-t004:** Run time comparison of optimization by the actual model and by Kriging.

Function Name	Gamultiobj	Paretosearch
Number of solutions	70	60
Regional Kriging time (s)	$8.62\times 10^{3}$	$7.03\times 10^{2}$
Simulation model time (s)	$1.59\times 10^{5}$	$3.44\times 10^{4}$

Regional Kriging effectively shortens optimization time.

**Table 5 entropy-22-00527-t005:** The TOPSIS score of selected Pareto solutions.

$K_{\mathrm{Pl}}$	$K_{\mathrm{Il}}$	$K_{\mathrm{Dl}}$	$K_{\mathrm{Pf}}$	$K_{\mathrm{If}}$	$K_{\mathrm{Df}}$	Score ($10^{-1}$)	Source ${}^{a}$
0.300	0.0001	0.300	0.291	0.164	0.145	2.295	regional Kriging
0.191	0.0001	0.300	0.290	0.0001	0.300	2.286	regional Kriging
0.211	0.042	0.173	0.089	0.136	0.286	1.324	simulation model
0.019	0.0001	0.131	0.122	0.0009	0.131	1.286	simulation model
0.300	0.132	0.263	0.254	0.0009	0.263	1.121	simulation model
0.299	0.014	0.070	0.117	0.300	0.145	0.898	regional Kriging
0.299	0.014	0.300	0.117	0.300	0.145	0.789	regional Kriging

${}^{a}$ Regional Kriging: regional Kriging gets the solution; simulation model: the simulation model gets the solution. The Pareto solutions of regional Kriging are reliable, and regional Kriging successfully replaces the simulation model.

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
