# Peer review of "Rapidly Tuning the PID Controller Based on the Regional Surrogate Model Technique in the UAV Formation"

_entropy, 2020, doi:10.3390/e22050527_

Round 1

Reviewer 1 Report

Observations:

- Section 1.Introduction = OK;

- In all paper: Put references for eqs. and tables (when is necessary)!

- P.2, R.62; P.13, R. 292: Replace „RMST” with „RSMT”!

- P.3: Fig.1: Why don’t use 0XYZ axes? Unmanned aerial vehicle moved on three axes!

- P.6, R.156: Why are you choosing this values: a = -5, b = 5, xmin = -1000, and xmax = -1000 (???)?

- P.9, R.213-214: Why are you choosing those values?

- P.10, Table 1; P.11, Table 2: Why k modify between: [0,0.1] to [0,0.6]?

- P.11, R.254: „We try to tune two PID controllers whose six inputs all belong to [0, 0.3].” Why?

- P.12, Fig.7: Clearly specify what represent symbols (blue and red) from this figure?

- All references are in text.

Reviewer 2 Report

Authors propose a leade following an approach based on PID controllers, which are tunned considering the regional information entropy. The proposal is quite interesting and sound. Nevertheless, there are some points that require attention. 

1.-There are too many acronyms in the abstract, which become difficult to follow. 

2. Some recent literature is missing:

https://doi.org/10.1007/s10846-015-0295-y

10.1109/LRA.2019.2901683

https://doi.org/10.1177/0020294018824764

https://doi.org/10.3390/math7111090

https://doi.org/10.3390/mca23040060

3.-I'm reading page 3, and here, I did'not understand what type of UAV are authors dealing? Is this a quadrotor or a fixed-wind?.

4.-The mathematical model looks quite simple. The control inputs are given for velocities instead of trust and forces. On the other hand, I don't see anywhere the effect of gravity.

5.- Include a block diagram of this:

"The 98 UAV formation is divided into the outer loop and the inner loop, which contain PID controllers 99 and linear quadratic regulator (LQR) controllers, respectively. The outer loop controls the position 100 dynamics to maintain the desired formation; the inner loop controls the UAV itself "

6.- Only two PID controllers?, is this approach valid only for a leader an a follower?

7.- It is not clear how to obtain (4) and (5).

8.- Why there are many errors in the model prediction

9.-Include noise in the simulation

10- I would like to see a plot, for example, by following a sinusoidal reference. 

Reviewer 3 Report

The introduction section should be further improved.

Literature is poor, please add the details about the latest papers related to the field.

Compare the proposed method with the other relevant methods

Please include the ROC, sensitivity and specificity for classification results

Improve the discussion and conclusion section.

Round 2

Reviewer 2 Report

Authors have addressed all my concerns. I don't have more comments. 

Reviewer 3 Report

The authors have addressed all my comments satisfactorily and the paper can be considered for publication.

This manuscript is a resubmission of an earlier submission. The following is a list of the peer review reports and author responses from that submission.